# Developing Co-Creation Research in Food Retail Environments: A Descriptive Case Study of a Healthy Supermarket Initiative in Regional Victoria, Australia

**DOI:** 10.3390/ijerph20126077

**Published:** 2023-06-07

**Authors:** Carmen Vargas, Jillian Whelan, Louise Feery, Deborah Greenslade, Melissa Farrington, Julie Brimblecombe, Freddy Thuruthikattu, Steven Allender

**Affiliations:** 1Global Centre for Preventive Health and Nutrition (GLOBE), Institute for Health Transformation, School of Health and Social Development, Deakin University, Geelong, VIC 3220, Australia; jill.whelan@deakin.edu.au (J.W.);; 2Ballarat Community Health, Ballarat, VIC 3350, Australia; louisef@bchc.org.au (L.F.); deborahg@bchc.org.au (D.G.); melissaf@bchc.org.au (M.F.); 3Department of Nutrition, Dietetics and Food, School of Clinical Sciences, Monash University, Clayton, VIC 3168, Australia; julie.brimblecombe@monash.edu.au; 4Primary Care Connect, Shepparton, VIC 3630, Australia; fthuruthikattu@primarycareconnect.com.au

**Keywords:** food environments, food retail, co-creation, health promotion, frameworks, healthy environments

## Abstract

Research into the co-creation of healthy food retail is in its early stages. One way to advance co-creation research is to explore and understand how co-creation was applied in developing, implementing, and evaluating a heath-enabling initiative in a supermarket in regional Victoria, Australia. A case study design was used to explore and understand how co-creation was applied in the *Eat Well, Feel Good Ballarat* project. Six documents and reports related to the *Eat Well, Feel Good Ballarat* project were analyzed with findings from the focus groups and interviews. Motivations to develop or implement health-enabling supermarket initiatives differed among the participants. Participants considered that initial negotiations were insufficient to keep the momentum going and to propose the value to the retailers to scale up the project. Presenting community-identified needs to the supermarket helped gain the retailer’s attention, whilst the co-design process helped the implementation. Showcasing the project to the community through media exposure kept the supermarket interested. Retailers’ time constraints and staff turnover were considered significant barriers to partnership building. This case study contributes insights into applying co-creation to health-enabling strategies in food retail outlets using two co-creation frameworks.

## 1. Introduction

Weight gain and poor health outcomes have been consistently related to food choices and dietary patterns [1,2]. Therefore, food retail environments are emerging as a key target for public health initiatives. Food choices tend to be made with little or no conscious health awareness [3,4] and are dictated by features of the environment, such as the type of food available [5,6]. Researchers and public health professionals are interested in changing food environments to support healthier food purchases. Supermarkets and food stores [6,7,8,9] are key because the proportion of foods purchased from supermarkets and grocery stores indicates a significant contribution to food intake (e.g., Europe 70–80% [10], USA 74% [11], and Australia 66% [12]), [13,14,15,16,17] and makes them strategic settings for health-enabling initiatives [5,6,15,16,17,18,19].

Multi-faceted interventions within supermarkets and grocery stores can improve the nutritional quality of food purchases and population health [20,21,22,23]. Interventions in supermarkets/grocery stores typically seek to improve dietary behavior at the point of choice in food stores [23,24,25], though these are not always sustainable over the long term [23,24,26]. Successful healthy food outlet initiatives require the participation of store owners and managers [7,20,26], consumers [20], and support retailers in the implementation [7,26]. This mirrors the United Nations Sustainable Development Goals, which urge, among others, the principles of multisectoral action to be applied to maximize prevention [27]. This goal is aspirational; achieving collaboration with multiple stakeholders in designing, implementing, and measuring health-enabling initiatives in supermarkets and grocery stores is understudied. 

Co-creation provides one avenue to achieve this goal. Co-creation refers to the collaborative approach of creative problem-solving between diverse stakeholders, from problem identification and solution generation to implementation and evaluation [28,29,30]. It is characterized by initiatives where actors with different knowledge and experiences collaborate (e.g., evidence, lived experience) and use their resources and abilities to solve a shared problem [31]. For healthy food retail, co-creation may enable retailers, researchers, consumers, and other interested parties to construct a shared goal that facilitates the design and implementation of healthy food retail interventions [26].

There is limited peer-reviewed literature on utilizing co-creation concepts [28,32] in food retail environments [26,31]. Vargas et al. [29] presented a co-creation framework for developing public health initiatives, and the CO-creation and evaluation of food environments to Advance Community Health (COACH) framework provides a specific guide to the use of co-creation to improve the healthiness of food environments in practice [33]. In this study, we apply the COACH framework alongside a generic co-creation framework [29] to examine the extent of co-creation used in developing, implementing, and evaluating a heath-enabling initiative in a regional supermarket. The case study set out to answer the following research questions: 

As aligned to the co-creation frameworks, at what stage did stakeholders become involved with the *Eat Well, Feel Good Ballarat* project?In what ways can the features/steps/structures/processes of phase one of the *Eat Well, Feel Good Ballarat* project inform co-creation frameworks?What is the stakeholders’ extent of participation and willingness to co-create health-enabling supermarkets?

## 2. Materials and Methods

### 2.1. Study Design

An instrumental case study design [34] was used to provide insight into how co-creation was applied in the *Eat Well, Feel Good Ballarat* project [35]. This instrumental case study uses Rosenberg and Yates’s [36] approach (Figure 1). It intends to provide an extensive and detailed understanding of co-creation theory and its impact on improving existing co-creation frameworks to develop health-enabling food retail initiatives [34,35]. All elements of the Standards for Reporting Qualitative Research (SRQR) guidelines [37] are included in the reporting of the study design, results, and analysis.

### 2.2. Context

We used a convenience case study selection to retrospectively apply the COACH and generic co-creation frameworks used in public health. The *Eat Well, Feel Good Ballarat* (EWFGB) project was conducted with supermarkets in the City of Ballarat, a regional centre located 110 km northwest of Melbourne in Victoria, Australia [38]. The City of Ballarat covers an area of 739 square kilometers, and in 2021 it had a population of 116,201 residents [39]. In 2020, Community Health Service (CHS) developed the *Eat Well, Feel Good Ballarat* (EWFGB) initiative in response to a community consultation where customers expressed the need for more supermarket support to choose healthier food and drink options [40]. 

The EWFGB project aimed to increase the ease for customers to identify and select healthier food and drink products using interventions within the supermarket environment that promote healthier food and drink options. This promotion was carried out using the Health Star Rating (HSR) system (a national front-of-pack labelling system that rates the overall nutritional profile of packaged food and assigns it a rating from ½ a star to 5 stars) [41] and a health promotion campaign [40,42].

### 2.3. Underpinning Theory

Co-creation highlights the importance of stakeholders’ interactions as the locus of the creation of value propositions [43]. In this view, value can be co-created between stakeholders by facilitating experience-based interactions that benefit all stakeholders [43,44,45,46,47]. Co-creation aligns with participatory research [43], engaging stakeholders in co-design or co-production approaches [44] and enabling those stakeholders to construct a shared agenda that facilitates collective action and creating valuable solutions [45]. 

### 2.4. Data Collection

Data were collected using multiple sources of evidence document review, focus groups, and interviews to promote the rigor of the case study description [34]. The document review comprised internal unpublished documents related to the EWFGB project provided by CHS (i.e., project plan, communication plan, evaluation report, and volunteer manual). Focus group and semi-structured interviews were conducted by the first author using an interview schedule. The schedule was developed post the document analysis based on elements from the co-creation theory. It was designed to explore aspects of the co-creation frameworks used for analysis that were not fully developed in the provided reports. The interview schedule also sought to capture participants’ experiences related to the project development, implementation, and evaluation (Appendix A). 

The schedule used in the focus group was refined and condensed to fit a 30 min one-on-one semi-structured interview. The focus group and interviews were conducted in English and audio-recorded via Zoom [48] (as the participants’ preferred option) at an agreed time between September-October 2022. All recordings were transcribed, de-identified, and cross-checked against the recordings. A copy of the interview transcript was returned to the participants for editing and checking [49]. Minor edits were made to the focus group transcript, and no modifications were made to the interviews. 

### 2.5. Data Analysis

#### 2.5.1. Document Analysis

Document analysis [50,51] was carried out by the first author on all documents provided by CHS. This process involved arranging data in chronological order and writing up the data according to a generic co-creation public health approach (initiation, identification, definition, design, implementation, and evaluation) [29]. Once the information was organized, the co-creation frameworks were used to summarize the data. The analysis was cross-checked by a second author (J.W.) and the CHS corroboration. The focus group and interviews helped interpret the document review and the application of the co-creation framework.

#### 2.5.2. Thematic Analysis

Thematic analysis [52,53] of the focus group and interviews was used to identify the perceived implications of co-creation to develop health-enabling initiatives in supermarkets and learnings and expectations for future implementation. The coding was completed using deductive and inductive thematic analysis [52]. For the inductive analysis, codes were identified from the “bottom-up”, deriving codes and themes from the data. Concepts of the Motivations, Opportunities, and Abilities model informed the deductive analysis. Common themes were identified by comparing and contrasting the coded data from each transcript [52]. One research team member (J.W.) reviewed the overall coding of data and themes. All coding and theme development were completed using NVivo 12 Software (Lumivero, Denver, CO, USA) [54]. Themes were synthesized narratively and conceptualized into the MOA model (motivations, opportunities, and abilities).

### 2.6. Analytical Filters for Data Analysis

#### 2.6.1. Co-Creation Frameworks

Two frameworks were used to analyze the extent of co-creation use in developing, implementing, and evaluating the EWFGB project in a regional supermarket. To date, specific implementation guidance for food environment change is limited. The multi-layered relationship between individuals/populations and their environments calls for frameworks to ideate initiatives that could better understand the dynamic behavior of this complex system and effectively improve the food retail environment for various societal outcomes and transdisciplinary collaboration [18,55]. 

These frameworks (COACH and generic co-creation in public health) have similar components and considerations (Appendix A) and consider the complexity of this setting. Both co-creation frameworks include multiple stakeholder inputs throughout the initiative, including development, implementation, evaluation, and sustainment. Some differences are observed in specific components. For example, the COACH framework highlights communications more explicitly than the generic co-creation framework in public health; the latter incorporates active communication in each stage. Additionally, the COACH framework focuses less on the stakeholders’ roles and more prominence on policy change and/or alignment than the generic co-creation framework in public health. The COACH framework provides a checklist to guide its application [33]. The research team adapted this checklist to reflect the tasks that should be considered in each stage of the co-creation frameworks. It was used to categorize and display the document review data.

Both frameworks entail continual improvement of outputs or outcomes as an incremental change and transformative innovation instead of a sole initiative. These are the main points of difference from other health promotion planning frameworks (e.g., Predisposing, Reinforcing, and Enabling Constructs in Educational Diagnosis and Evaluation - Policy, Regulatory, and Organizational Constructs in Educational and Environmental Development [PRECEDE-PROCEED] [56], Intervention Mapping [57], or Victorian Department of Health Integrated Health Promotion Planning Framework [58]).

#### 2.6.2. Motivations, Opportunities, and Abilities Model

The MOA model was used to explore the participants’ participation and willingness to co-create health-enabling supermarket initiatives. Identifying stakeholders’ motivations, opportunities, and abilities (MOA) can help to target strategies that enable readiness to co-create [59,60,61]. Table 1 shows the definitions of these three concepts and the assumptions in the context of this case study.

### 2.7. Data Reduction and Display

Data were reduced to three points to present the extent of co-creation use in developing, implementing, and evaluating a heath-enabling initiative in a regional supermarket. (1) Overview of the stakeholders’ stage of involvement in the EWFGB project according to each co-creation framework; (2) application of the adapted COACH checklist; and (3) Overview of qualitative data and theme description.

### 2.8. Research Reflexivity Statement

The research team was strongly committed to working collaboratively, collecting, and analysing data from the study’s inception. Individual involvement varied in diverse stages of the process. Two team members most closely involved in the process (C.V., J.W.) frequently met (at least once a week) to discuss the progress of fieldwork and reflect on data collection. During the early stages of analysis, two team members (C.V., J.W.) repeatedly met to align the data extraction tool (COACH checklist).

An ongoing dialogue between the researchers and Community Health Service (CHS) was developed using e-mail contact and online meetings. Meetings with CHS were held more frequently in the early planning stages, and feedback was requested during data analysis and publication writing. These checks were conducted to verify the accuracy and credibility of the interpretations. At the final analysis stage, input was sought from other research team members with extensive experience conducting qualitative studies and implementing complex initiatives in food environments (J.B., S.A.). This work culminated in a retrospective instrumental case study informing the COACH framework and the application of a generic co-creation health promotion framework to food retail outlets. 

### 2.9. Ethical Considerations

Ethics approval was obtained from the Deakin University Faculty of Health (HEAG-H 63_2021). All participants provided informed consent to provide documents for analysis, for focus groups and interviews to be recorded, and to use direct quotes in a non-identifiable form.

## 3. Results

Six different documents and reports related to the EWFGB project were provided and analyzed. A full description of the *Eat Well, Feel Good* Ballarat project was developed as a case study by consolidating these documents and is presented as an additional file. The results are presented in three parts: Section 3.1: application of the features, structures, and processes of the EWFGB project according to each co-creation framework; Section 3.2: overview of the stakeholders’ stage of involvement in the EWFGB project according to each co-creation framework; and Section 3.3: report of focus group and interview implications for co-creation health-enabling initiatives in supermarkets.

### 3.1. Checklist Application

Table 2 outlines the phases of two co-creation theoretical approaches, the required tasks to be considered in each stage, and the EWFGB project alignment as application examples.

### 3.2. Stakeholders’ Stage of Involvement

Community Health Service was the principal organization working with the supermarket’s management team on the EWFGB project since 2019. Multiple stakeholders were involved in different stages of the co-creation process (Table 3).

### 3.3. Implications of Co-Creation

One focus group with four CHS staff members and two interviews (one former CHS health promotion officer and one supermarket manager) were conducted. Nine themes were identified across the three theoretical dimensions. Figure 2 depicts an overview of the data structure. Each theme is described in more detail below.

#### 3.3.1. Motivations

Comments showed a mix of intrinsic (i.e., improving community health) and extrinsic (i.e., utilitarian) motivations to co-create more health-enabling strategies in food retail outlets. Motivations differed by the participants’ organization; for example, while health promotion participants focused on community benefits, sales and profit were the retailer-participant’s primary motivations.

*Intrinsic motivation: I think it’s really important to keep going, but just be mindful of what’s making an impact [within the community]*.(FGP)

*Extrinsic motivation: […] as a business, you will always probably try and lean towards where you’re going to get the most of your sales from and the most of your profit to keep your business viable. […]*.(RI)

There was an acknowledgement and value in supermarkets’ efforts to stock healthier products. For instance, a participant reflected that:

*[…] nowadays, compared to the last five years, we can always see new product availability […] you can see sections of healthier food items coming up in place, you can see all supermarkets focusing on putting vegetables and fruits right in front of the stores*.(HPI)

Furthermore, the conviction that independent and big chain supermarkets have the motivation to act in the health space, but that how they are putting it out is different from what health practitioners would have done or expected.

*What we’re trying to change […] I guess, is that everything that we sort of make in-stores, we might have pre-cut fruit, we do our baking of healthy bread and all that stuff, is to try and get any department or positions in store and try and change that consumer shopping habit [to a healthier one]*.(RI)

The EWFGB project did not get initial traction from a big chain of supermarkets. Participants’ reflections relate to the idea that local independent supermarkets would like to implement health-enabling strategies in their stores for the community. One participant narrated the initial successful conversation:

*I contacted [Retailer’s name], who manages six supermarkets in our catchment […] I just rang the general number, and I said: “can I speak to the owner of the business?” And she goes, “hang on, I’ll just put you through to [Retailer’s name],” it was so easy. I got through, and (you have that sort of elevated pitch to get his attention straight up) I spoke about the [Eat Well @IGA] project, which he knew about, and whether there was any opportunity to have some discussions with him and his managers about doing some work with them around some healthy initiatives. And he said, “yeah, sure, let’s make time now.” How lucky was that*? (FGP)

While all staff members across the supermarket chain did not share this initial motivation, it helped to initiate the EWFGB project. Yet it has not been enough to keep the momentum going and propose a sufficient value for retailers to scale up the project.

*I would say that I can see that if we look back over that particular store manager that really did get behind and more enthusiastic, and then others are doing it because [Retailer’s name] wants to do it. […] it’s still okay, but they might be just ticking through the motion […] they’re not personally invested in it […] It’s sort of an extra thing*.(FGP)

#### 3.3.2. Opportunities

Participants reflected on factors that reduced the barriers throughout the EWFGB project. Most of these factors relate to CHS’s understanding of the supermarket’s extrinsic motivations for involvement in the project and the work done on relationship building. 

*There were quite a lot of planning meetings in the initial stages, and we went in there very cautiously because everything we [had] read was about how to maintain their profit, and of course, they do, but they were just really trying to be very helpful (FGP). I feel like it’s a new project [EWFGB], so they would want to actually put that project in the supermarket because, at the end of the day, the project means more promotion, more publicity and more people spending more time in the stores and therefore more profit*.(HPI)

The co-design process facilitated negotiation to balance the community-identified needs and the supermarket’s possibilities. Showcasing the project to the community through the supermarket’s media exposure was very successful. Financial constraints determined the extent and type of implemented strategies. Optimizing materials (e.g., quality, design, and types of promotional material) was also an essential factor for the sustainment of the project (i.e., materials being durable). For the retailer participant, the determination of roles where CHS took most responsibilities of the project and the project’s administration was highly recognized.

*We [CHS and Supermarket] set it all up, we had some teams meeting to start with, and we had follow-up meetings in-store, which is great. All the point-of-sale material was presented and signed off, making sure that it was all suitable at a store level. That was a great initiative*. *[…]*.(RI)

Participants’ learning from this implementation led to sharing their reflections for future practice. Broad participants’ comments related to supporting local producers to foster a sense of community. Internal or national policy to restrict the promotion of unhealthy products was also supported. Insights for implementation considered the implementation of healthy nudges, using store resources for promotion (e.g., screens or store audio), or involving other stakeholders (e.g., community, students) could support the implementation of the strategy and/or make supermarkets accountable for their product displays. The most relevant for the health promotion participants related to sharing responsibilities (e.g., financial, staff, time) as supermarkets should be empowered to sustain and scale up this type of initiative. Currently, the supermarket is collaborating, so the project is implemented as the store’s capability and staff’s time allowed, but the retailers have not yet engaged as active partners. The retailer participant acknowledged this by saying:

*[…] if it was up to us to run it [the initiative], we probably wouldn’t do it because we just don’t physically have the time to do all this type of stuff with every other initiative we’ve got going on in the business*.(RI)

#### 3.3.3. Ability

While active collaboration and strong partnerships between stakeholders are essential for the success of a health promotion initiative, the supermarket’s staff turnover, changes in store management, and new commitments (i.e., new store opening) reduce the chances for ongoing supermarket engagement and project ownership. Resources (i.e., time and staff) that need to be constantly invested in maintaining this working relationship with the supermarket are unfeasible for the health organization. 

*The situation is that we’re constantly trying to re-connect and re-establish that relationship with the supermarket […] trying to just continue that kind of momentum, [which] it’s a little bit tricky at this end of the project*.(FGP)

In the project planning stage, the definition of roles was established. The supermarket’s directions dictated the allocation of responsibilities, which led to a relationship disconnection. The CHS staff tried to be helpful, were willing to work in this setting, and respected the supermarket’s knowledge and space.

*There was just a lot of work around working harmoniously together to get the best results for the EWFGB but also making sure that for the supermarket, it worked well with their business as well. […] it has to be some give and take because a lot of our work too; how are we going to sustain it? We’ve put so much effort into keeping the marketing collateral up; that does take a lot of resources. […] including training of staff to be included in their orientation and induction training as well. That’s what we would really love. But it hasn’t been [happening], as far as I know*.(FGP)

While there was an initial commitment to implement the EWFGB project, CHS has had to be more flexible in adapting its strategies, which has led to a commitment imbalance for the project itself. Despite the extra resources allocated by CHS in the form of staff and materials (e.g., during implementation and evaluation), the time allocated to the project seems to be a significant stress point for the supermarket. 

*[…] I think in the first instance, it always comes down to how much time we can actually provide or allocate to a project like this, and if you’ve got the support of Ballarat Health, they can do a lot of the groundwork, it just makes it easy for us to sign stuff off and say, “yeah, we could do it” [...] any sort of training videos that they’ve done they add on the top, so you’ve got to get people to be able to find the time to actually sit there for a five-minute session […] it can be a bit frustrating or a problem to watch something, to learn something about the healthy eating concept, that is one of the things that [is an issue] […] I think there’d be a commitment there, but it’s just that [it’s difficult]. […] I think it’d be a commitment to take it further, but it’s just the amount of time that we would have available to actually get into a deeper process or where we head with it*.(RI)

The EWFGB project had some implementation barriers beyond their control, such as COVID-19 lockdowns and chain supply issues related to the floods. These barriers are difficult to predict. Other barriers relate to the competing messaging in-store with a high seasonal thematic product promotion (e.g., Easter, Halloween, or Christmas). Additionally, the supermarket has financial commitments that do not allow for a sustained health-enabling strategy. 

*There’s a lot of stuff they wanted to do this year that they couldn’t do because we had some trading terms agreements with certain suppliers […], and they [CHS] wanted to get fresher products in certain positions in the store, and we weren’t able to do that because of […] those terms that we had to some key supplies which makes it really difficult to sort of implement everything that needed to be implemented*.(RI)

## 4. Discussion

The co-creation was found to be a suitable approach to designing, implementing, and evaluating initiatives within food retail outlets. Using a co-creation approach to explore the features, structures, and processes of health-enabling initiatives in food retail outlets leads to a better understanding of the dynamic behavior of this complex system. 

The co-creation approach is not a linear process; it has many components happening in parallel that are continuously reinforced and may feed other features within the process (e.g., governance or communication) [45,64]. The COACH framework emphasizes the importance of communication within the co-creation process. Our results showed the relevance of developing a communication plan for the multi-stakeholder relationship-building process. The communication plan could be an engagement tool that considers elements that facilitate collaborative, dynamic, contextual, and generative interactions. These characteristics, in turn, generate mutual value propositions through productive and meaningful experiences [46,65].

The EWFGB project ensured processes for internal communications (i.e., volunteers’ manual), as well as to report results to the supermarket and disseminate efforts to the community (i.e., local media coverage). Yet, engagement platforms (i.e., processes, interfaces) to help stakeholders interact and share experiences between them and with CHS were limited. The benefits of transparent, trusting, and open communication are central to the co-creation of value, as purposeful interactions among multiple stakeholders enable the creation and extraction of value [30,66], which in turn can promote innovation and sustain change [45,64]. 

Our results showed the EWFGB project’s efforts to establish clear roles and accountability measures for the teams controlled by CHS (e.g., volunteer groups and university students). Yet expectations for the retail party that could lead to implementation accountability and commitment were flexible, leaving this stakeholder group uncommitted in an equal capacity (e.g., staff, time, or funds). Clear roles and expectations must be agreed upon and included in the communications plan and processes by which engagement experiences could be expanded together [30,66]. Learnings from Healthy Stores 2020 [9] showed that clarity in stakeholder roles and expectations can maintain long-lasting relationships and can lead to internal policy change. 

CHS coordination was instrumental in successfully implementing, maintaining, evaluating, and continuing the EWFGB project; top-down governance tends to provide significant inputs from only one stakeholder towards their goal. Our results also showed that the partnership remained in two independent silos (public health and food retail), where the supermarkets collaborated with CHS by supporting the initiative. Multiple research studies have identified governance as necessary for success, as it provides coordination, guidance, impact, and vision for the work [67]. Interdependent horizontal partnership relations are an enhanced mode of governance in which plans, strategies, and policies are developed, transforming the hierarchical relations between public and private actors [45]. Research has shown that an ad hoc working group is one way to build horizontal and cross-sectoral collaboration [68].

Since improving food retail environments in collaboration with retailers is a relatively new approach, it is common to see academic researchers [69] or organizations such as CHS leading the implementation and providing the funding. It could be theorized that figures/organizations holding the funds to conduct health-enabling strategies in food retail outlets will assume the leadership role, causing a power imbalance in this multi-stakeholder relationship. Equal partnership between stakeholders will ensure stakeholder empowerment, project appropriation, and value co-creation [70]. Supermarkets are expected to become actively engaged in the co-creation of value, requiring them to commit resources, time, and energy to the co-creation process [45]. Including internal policies that could support affordable healthy options for consumers and reduce price promotion on unhealthy products.

Our results did not provide sufficient evidence of motivators that could lead supermarkets to sustain and scale up the EWFGB project by themselves, such as perspectives towards the feasibility of an internal policy change that could potentially improve the healthiness of the supermarket. The retailer’s buy-in on the project showed the importance of implementing health-enabling strategies in supermarkets, which agrees with several studies that have demonstrated the impact of retailers on the successful implementation of strategies that have not caused economic loss [7,20,26,69]. Understanding factors that affect supermarkets’ decision to improve the food environment will facilitate an effective and long-term healthy food policy and initiative development [71]. 

Our results showed that supermarkets’ time constraints and staff turnover are the central issues in active collaboration, initiative engagement, and ownership. Research has shown that the extra workload added by a health-enabling strategy is an important point of resistance for retailers to implement them [72]. Time is essential for co-creation to be successful, as opportunity identification, problem analysis, and solution development happen through an ongoing process of information and ideation sharing [30]. This process yields important information and ideas that could improve outcomes as stakeholders provide different and significant input over time [30,45]. In this sense, if stakeholders are not prepared to give the time that a co-creation process requires, it may not be the right approach. 

The co-creation approach calls for the ongoing involvement of new stakeholders, as these will bring new perspectives and solutions [28]. The results of this study suggested that involving other stakeholder groups, such as students or volunteers, could help to mitigate the extra workload that a health-enabling strategy in a food retail outlet could add. Additionally, involving consumers in the mix of stakeholders can help to leverage supermarkets’ time investment and accountability. Previous research has shown that involving consumers in the design of strategies creates a higher level of interest and commitment from the retailer to implement those strategies [8,9].

### 4.1. Strengths and Limitations

While the EWFGB project was not conceived through a co-creation process, it has helped to inform essential co-creation attributes (i.e., active collaboration, interactions, and value co-creation) through real-practice examples in supermarkets. The EWFGB project provides clear examples of almost all components of both co-creation frameworks, which could help the future application of co-creation as an approach to health-enabling strategies in food retail outlets. The systematic application of co-creation to develop health-enabling food retail strategies is understudied; our study provides examples that could be applied to almost all components of two co-creation frameworks, which could help for future planning. A weakness of this study lies in the lack of retailers’ voices, which could have provided a deeper understanding of the motivations and value propositions that could be considered in future health-enabling initiatives.

### 4.2. Implication for Research and Practice

Conditions that support or hinder the successful implementation of co-creation must be carefully identified and examined to fully exploit co-creation as a fruitful way to tackle challenges [28,45]. Co-creation could be used to develop a local food retail environment policy—one that seeks to acknowledge ‘value’ from all parties. Learnings from this case study could be applied to the co-creation of health-enabling strategies in food retail outlets. Involving consumers in a meaningful way in other co-creation stages besides the evaluation should be considered as leveraging points to continual retailer engagement, participation, and commitment to ongoing implementation. Future research should focus on capturing the views of other essential stakeholders (i.e., food industry, transportation, retailers, consumers, suppliers) to complete the multi-stakeholder ecosystem that can inform and advance co-creation research. Additionally, these two frameworks could be trialed independently to reduce the implementation gap between the learnings of this retrospective case study and a strategy that uses a co-creation approach from the initial stage. Furthermore, given the complexity of food retail outlets, there is the need to develop and test health-enabling strategies that can be adapted to the expected (e.g., in-store promotion) or unexpected (e.g., floods, COVID-19) barriers. 

## 5. Conclusions

Co-creation is a time-consuming process, which lowers the approach’s feasibility. Yet, involving other stakeholders could counterbalance this barrier. The lack of views from different stakeholders limited our understanding of how to foster meaningful collaboration and practical suggestions to promote partnerships in this setting. This case study contributes to a better understanding of applying co-creation to health-enabling strategies in food retail outlets using two co-creation frameworks. It identifies opportunities for co-creation to develop, implement, and evaluate future health-enabling initiatives in food retail environments.

## Figures and Tables

**Figure 1 ijerph-20-06077-f001:**
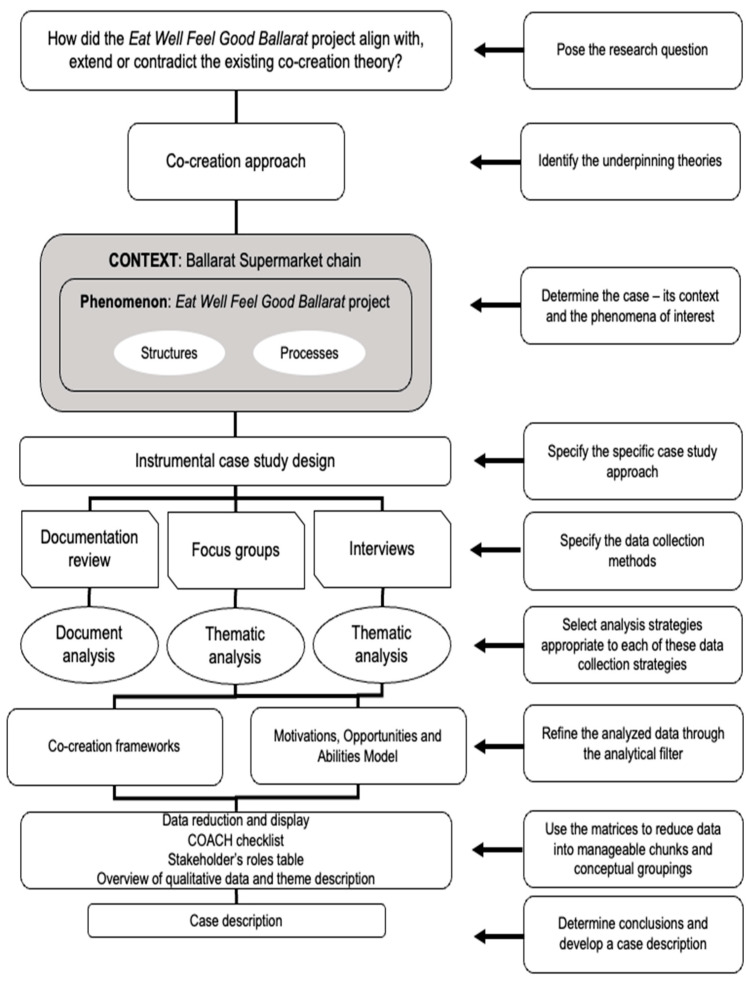
Case study design.

**Figure 2 ijerph-20-06077-f002:**
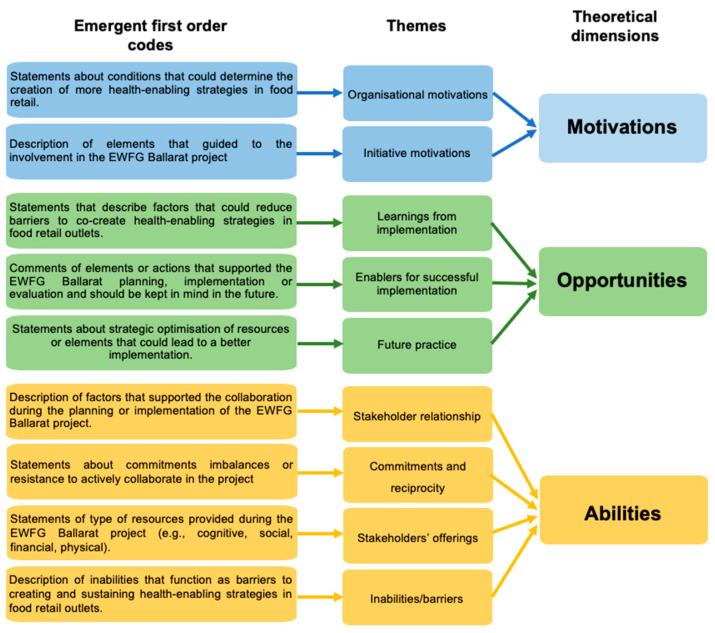
Overview of data structure.

**Table 1 ijerph-20-06077-t001:** Motivations, Opportunities, and Abilities Model.

	Definition	Assumption
Motivations	The force that directs individuals towards goals. It reflects readiness and interest to engage in an activity [62].	Essential condition to co-create more health-enabling strategies in food retail. The goals are not restricted to the initiative’s short- and long-term objectives but could extend to the organizational motivations to achieve change in the food outlet.
Opportunities	Relevant constraints that enable the desired outcome [62]. Opportunities could extend to situational and operational factors that support or serve as barriers to performing an activity [60].	Circumstances that enabled the EWFGB project could lead to its long-term sustainability. The opportunity could reflect actions/strategies/structures that increase the means or reduce barriers to co-create health-enabling strategies in food retail outlets.
Abilities	The necessary resource level and the extent of these resources to achieve the desired goal [59]. For co-creation, this relates to the stakeholders’ skills or knowledge to engage in co-creation (e.g., platforms and skills to interact with others and capabilities to exchange value propositions) [61].	Type of resources (e.g., cognitive, social, financial, physical) the stakeholders enable the co-creation of health-enabling strategies in food retail outlets.

**Table 2 ijerph-20-06077-t002:** Co-creation phases and application of the *Eat Well, Feel Good* Ballarat project.

COACH	Generic Co-Creation	Task	Application of the *Eat Well, Feel Good Ballarat* Project
Stakeholder engagement, evidence collection and governance	Identify	Identify the governance/management arrangements and organizational structure supporting, influencing, and resourcing relevant to the issue of interest	CHS is the principal organization that has closely partnered with the Supermarket since 2019 [42]. Partners: Supermarket and CHS. A supermarket owner is highly engaged and has an altruistic motivation to help the community [42]. A Supermarket in Ballarat has agreed to partner with CHS on the pilot [42].
		Identify and analyze motivation for change	Results from a community consultation [40]. Ongoing work in a Supermarket in Bendigo [7].
		Identify any existing tools to measure and assess change in the food environment (consider if these are appropriately validated and reliable)	Store Scout was used to audit the three supermarkets before implementing the project [42].
	Analyze	Analyze stakeholder network and role agreement	Supporters: Table 3
		Analyze key area(s) of concern identified by stakeholders	Customers wanted more support from supermarkets to help them make healthier food and drink choices [40]. Barriers to making healthy choices in supermarkets relate to the low cost of unhealthy foods, misleading and hard-to-understand labelling, and the layout of the stores [40].
		Identify any relevant implementation frameworks/strategies to be considered	The EWFGB project was modelled on key elements of Eat Well @ IGA [7].
	Define	Prioritize problems and possible solutions	Increase the ease for customers to identify and select healthier food and drink products using interventions within the supermarket environment that promote healthier food and drink options [40]. The Australian Government’s HSR system helps customers identify the healthiest food and drink options while shopping at the supermarket [41].
Communication, policy alignment & development		Identify any government/internal policy that needs to be implemented, aligned to or to be developed	The intervention sits within a broader movement of health promotion strategies [63].
		Ensure effective communication and information sharing among stakeholders (consider channels that enable continually feedback between stakeholders)	Regular meetings occur during the definition of actions and initiative development. The Supermarket had a direct communication line to report any issues with the materials.
		Identify and consider any relevant initiative dissemination strategies that benefit or enhance the initiative (e.g., communication targeted to communities)	Local newspapers and TV channels promoted the project to the community, social media, and CHS website [42].
		Ensure any privacy and/or ethics considerations (if relevant)	Privacy was ensured according to CHS guidelines. All participants in the evaluation provided informed consent.
Co-design of evidence-informed action and implementation planning	Design	Use methods to develop a shared understanding of the drivers of the issue and potential solutions to this issue (e.g., Group Model Building or other participatory methods)	NA
		Ensure that the prioritization of action/strategies was collaboratively defined	CHS developed insights from the literature and pilot studies. The included strategies were selected in agreement with the store manager. The supermarket provided additional input on the type of promotional materials. CHS adapted designs and evaluated the feasibility of implementation.
		Prepare a specific plan for the implementation of strategies (consider processes and timeline)	Categorization of key products by HSR Initial Store Scout audit. Developing and implementing specific in-store messaging (signage, planograms, recipe, weekly specials, etc.) Development and implementation of shelf signage to promote 4.5 and above. Development and delivery of staff training. Develop key communication messages.
		Construct a resource and asset allocation for the implementation of actions	CHS—EWFGB project management. Victorian State Government Department—Funding. University partners: information and provision of support. Volunteers: data collection. Supermarket—Planning and implementation support.
		Develop an initiative evaluation plan before starting the implementation	Evaluation process: surveys, interviews, and final Store Scout audit, sales data.
	Realize	Implement and monitor the initiative, identify trends and adapt if/when needed	Pre-intervention Store Scout analysis was undertaken between the 19th and 23rd of April 2021. Post-intervention Store Scout analysis was undertaken between the 17th and 21st of May 2021 [42]. The eight-week pilot was planned for each store, but this was extended to nine after a two-week state-wide lockdown between the 28th of May and 10th of June 2021 impacted store monitoring and the commencement of customer surveying. The launch had a staggered start (of one week) between stores so that process evaluation of the implementation could guide each subsequent launch [42].
Identify Momentum continuous quality improvement (CQI) cycle	Evaluate	Assess if/to what extent the proposed outcomes were met	EWFGB project demonstrated the capacity of the supermarket environment to support consumers in making healthier food and drink choices. More detail in the evaluation report [42].
		Identify the critical area (s) of concern (consider if a new concern/ problem has emerged)	CHS staff, volunteers, and interviewees were concerned that it might be hard to sustain the materials once this assistance is taken away. Managers noted that the duration of any marketing intervention requires careful consideration describing how customers can get “store blind” if promotional materials are used for an extended period.
		Identify encountered barriers that limited the implementation and/or stakeholder collaboration	Limited supermarket staff engagement due to staff capacity limitations, timing, and preparedness before scheduled implementation. HSR ratings of Supermarket products assessment were done in early 2020 but were used until early/mid-2021 (many products and ratings may have changed due to introducing new products, discontinuing products, and product reformulation). COVID-19 restrictions limited the ability to meet staff and hold meetings in person.
		Identify enablers that supported or enhanced the implementation and/or stakeholder collaboration	Visually attractive materials which help customers reflect on nutrition. Recipe cards were the most popular promotional material. The supermarket commitment to the trial and provision of the space and sales data.
		Learnings from the diverse stakeholders	Supermarket: The senior managers strongly supported the EWFGB project continuing in the future, recommending sustainability of the promotional materials as a consideration. Support from CHS staff and volunteers throughout the project was critical to the success of the project.
		Identification of new stakeholders	NA

**Table 3 ijerph-20-06077-t003:** Stakeholders’ stage of involvement in the *Eat Well, Feel Good* Ballarat project.

COACH	Stakeholder Engagement, Evidence Collection, and Governance			Co-Design of Evidence-Informed Action and Implementation Planning		Momentum and CQI
		Communication Policy Alignment & Development				
Generic Co-Creation	Identify	Analyze	Define	Design	Realize	Evaluate
Community Health Service	✓	✓	✓	✓	✓	✓
Regional Health Service	✓	✓	✓	✓		
Primary Care Partnership	✓	✓	✓			
Victorian State Government Department	✓	✓	✓			
Non-Government Organizations		✓	✓			
Academic Institution			✓	✓		
Universities	✓	✓			✓	✓
Supermarket			✓	✓	✓	✓
Community					✓	✓

## Data Availability

The interview transcripts analyzed during the current study are available from the corresponding author upon reasonable request.

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
