# Peer review of "Developing Co-Creation Research in Food Retail Environments: A Descriptive Case Study of a Healthy Supermarket Initiative in Regional Victoria, Australia"

_ijerph, 2023, doi:10.3390/ijerph20126077_

Round 1

Author Response

Thank you, Referee 1 for your comments. Please find attached our response.

Reviewer 2 Report

This research: (1) Well study design and complied with the good Standards for Reporting Qualitative Research (SRQR) guidelines; (2) EWFGB as a platform, to examine co-creation framework for developing, implementing and evaluating a heath-enabling initiative; (3) Practiced important values as equality and communication.

This research did mirror the United Nations Sustainable Development Goals, especially SDG 3: Establish Good Health and Well-Being, SDG 12: Influence Responsible Consumption and Production, and SDG 17: Build Partnerships for the Goals.

Based on sustainable development thinking:

1.     In reference 40, Greenslade et al. (Greenslade was also a co-author of this manuscript) mentioned that the barriers to making healthy choices in supermarkets: (1) low cost of unhealthy foods; (2) misleading and hard-to-understand labelling; (3) the layout of the stores.

From another viewpoint, the healthier food and drink products promoted by the EWFGB project, whether affordable to the general public, and the affordability might affect the sustainability of this project, especially the medium-term and long-term sustainability. Please make further discussion of this issue.

2.     Whether healthier food and drink products were affordable to general public, might affect the revenue of supermarket, which in turn affects the initiative of supermarkets to participate in this project, and makes it more difficult for supermarkets to sustain and scale up the EWFGB project. Please make further discussion of this issue.

Page 10 of 20, line 240-241:

“Ten themes” were identified across the three theoretical dimensions? There were nine themes in Figure 2. Please fix this error.

Author Response

Thank you, Referee 2, for your comments. Please find attached our response to your comments.

Reviewer 3 Report

In data collection section on page 4 check terminology.  The authors state that they use focus groups but then state that this process was revised and they used one to one semi-structured interviews.  So did the authors only use semi-structured interviews or do some focus groups take place? 

It may be worth describing in more detail as it is not clear who participated in the study.  For example a table of characteristics of the sample.  It would also then be helpful to go back to this in the discussion on how it may have influenced findings e.g. people need to be able to use Zoom.  (This is partially addressed by a mention in the weaknesses of lack of engagement of people in the retail sector. 

Author Response

Thank you, Referee 3, for your comments and suggestions. Please find attached our answer.
